

**Direct observations indicate photodegradable oxygenated VOCs**
**as larger contributors to radicals and ozone production in the**
**atmosphere**
Wenjie Wang[1,2], Bin Yuan[1,3]\*, Yuwen Peng[1,3], Hang Su[2]\*, Yafang Cheng[2], Suxia Yang[1,3],
Caihong Wu[1,3], Jipeng Qi[1,3], Fengxia Bao[2], Yibo Huangfu[1,3], Chaomin Wang[1,3],
Chenshuo Ye[1], Zelong Wang[1,3], Baolin Wang[4], Xinming Wang[5], Wei Song[5], Weiwei
Hu[5], Peng Cheng[6], Manni Zhu[1,3], Junyu Zheng[1,3], Min Shao[1,3]
[1] Institute for Environmental and Climate Research, Jinan University,
Guangzhou 511443, China
[2] Multiphase Chemistry Department, Max Planck Institute for Chemistry, Mainz
55128, Germany.
[3] Guangdong-Hongkong-Macau Joint Laboratory of Collaborative Innovation for
Environmental Quality, Guangzhou, 511443, China
[4] School of Environmental Science and Engineering, Qilu University of
Technology, Jinan 250353, China
[5] State Key Laboratory of Organic Geochemistry, Guangzhou Institute of
Geochemistry, Chinese Academy of Sciences, Guangzhou 510640, China
[6] Institute of Mass Spectrometry and Atmospheric Environment, Jinan
University, Guangzhou 510632, China
*Correspondence to:* Bin Yuan (byuan@jnu.edu.cn); Hang Su (h.su@mpic.de)



**Abstract:** Volatile organic compounds (VOCs) regulate atmospheric oxidation capacity,
and the reactions of VOCs are key in understanding ozone formation and its mitigation
strategies. When evaluating its impact, most previous studies did not fully consider the
role of oxygenated VOCs due to limitations of measurement technology. By using a
proton-transfer-reaction time-of-flight mass spectrometer (PTR-ToF-MS) combined
with gas chromatography mass spectrometer (GC-MS) technology, we are able to
quantify a large number of oxygenated VOCs in a representative urban environment in
southern China. Based on the new dataset, we find that non - formaldehyde (HCHO)
OVOCs can contribute large fractions (22-44%) of total $RO_X$ radical production,
comparable or larger than the contributions from nitrous acid and formaldehyde. We
demonstrate that constraints using OVOCs observations are essential in modeling
radical and ozone production, as modelled OVOCs can be substantially lower than
measurements, potentially due to primary emissions and/or missing secondary sources.
Our results show that models without OVOC constraints using ambient measurements
will underestimate $P(RO_X)$ and ozone production rate, and may also affect the
determination of sensitivity regime in ozone formation. Therefore, a thorough
quantification of photodegradable OVOCs species is in urgent need to understand
accurately the ozone chemistry and to develop effective control strategies.

**Keywords:** photolysis reactions; oxygenated volatile organic compounds; radical
production; ozone production



## 1 Introduction

Ground-level ozone is generated by photochemical oxidation of volatile organic compounds (VOCs) under the catalysis of nitrogen oxides (NOx) and hydroxide radicals (HOX=OH+HO2) (Atkinson, 2000;Monks et al., 2015). In this process, photolysis reactions are a crucial driving force. Photodegradable species (i.e. species that is capable of photolysis) including O3, nitrous acid (HONO), and oxygenated VOCs (OVOCs) can contribute to primary production of ROX (OH+HO2+RO2) radicals via photolysis reactions, thereby accelerating the recycling of radicals to generate ozone (Volkamer et al., 2010). The strong dependence of OH concentration on j(O$^1$D) was found in a number of field measurements (Ehhalt and Rohrer, 2000;Rohrer et al., 2014b;Stone et al., 2012), implying the dominant role of ultraviolet radiation and photolysis reactions in the production of HOX radicals. Edwards et al. (2014) found that the high ozone pollution in an oil and gas producing basin in the U.S. in winter was caused by the photolysis of high concentrations of OVOCs to generate sufficient oxidants. A recent model simulation with limited OVOCs measurements by Qu et al.(Qu et al., 2021) indicated that OVOC species is the largest free-radical source in the boundary layer. Another study indicated that fast ozone production during winter haze episodes in China was driven by HOX radicals derived from photolysis of formaldehyde (HCHO), overcoming radical titration induced by NOx emissions (Li et al., 2021). Therefore, an accurate quantification of numerous photolysis reactions is necessary to understand the mechanism of ROX radical and ozone production.

However, only limited number of photodegradable OVOCs species, such as formaldehyde, acetaldehyde and acetone, have been measured in the field campaigns in China due to the limitations of measurement technology (Lu et al., 2013;Lu et al., 2012;Tan et al., 2018;Tan et al., 2019c). Many important photodegradable OVOCs, such as larger aldehydes and ketones, carboxylic acids, nitrophenols, organic peroxides and multifunctional species, have been rarely quantified accurately in ambient environments. In such cases, the quantification of the primary production of ROX



radicals induced by photolysis reactions may not be adequately accurate. Many studies
used photochemical models to simulate unmeasured OVOC species (Tan et al.,
2019b;Volkamer et al., 2010;Ling et al., 2014;Edwards et al., 2014). However, large
uncertainties in the simulation of OVOCs remain due to primary emissions of OVOCs
(McDonald et al., 2018;Karl et al., 2018;Gkatzelis et al., 2021), missing secondary
sources (Bloss et al., 2005;Ji et al., 2017), heterogenous uptake of aerosols and
unknown dilution and transmission processes (Li et al., 2014). For instance, chamber
experiments of the oxidation of aromatics by OH radical indicated that MCM
mechanism generally underestimated the formation of aldehydes, ketones and phenols
by 10-70% (Bloss et al., 2005;Ji et al., 2017), implying the existence of unknown
production pathways for these OVOC species. Furthermore, model simulations
frequently underestimated observed $RO_X$ radicals in ambient studies of $RO_X$ radicals
(Hofzumahaus et al., 2009;Tan et al., 2018;Lelieveld et al., 2008;Rohrer et al.,
2014a;Sheehy et al., 2010;Emmerson et al., 2005;Ma et al., 2019). Given that only
limited photodegradable OVOCs species were measured in these studies, the lack of
comprehensive measurements of OVOCs to constrain the model is likely to be a cause
of the underestimation.
Thus far, the concrete effects of photodegradable OVOCs on radical and ozone
production remains unexplored in China. Based on comprehensive field observations
in a mega-city in southern China, a variety of important photodegradable OVOC
species were measured. The contributions of these photodegradable OVOCs species
to the production of $RO_X$ radicals are quantified, and the effect of photolysis reactions
on ozone production is quantitatively assessed.

## 2 Materials and Methods

### 2.1 OVOC measurements

Field measurements were conducted at an urban site in Guangzhou (113.2ºE, 23ºN)
from 14 September to 20 November 2018. The sampling site is located on the 9th floor
of a building on the campus of Guangzhou Institute of Geochemistry, Chinese Academy



of Sciences, 25 m above the ground level. This site is regarded as a typical urban site
in Guangzhou influenced by industrial and vehicle emissions.
During this campaign, an online PTR-ToF-MS (Ionicon Analytic GmbH,
Innsbruck, Austria) with $H_3O^+$ and $NO^+$ chemistry was used to measure ambient volatile
organic compounds (VOCs) (Wang et al., 2020a;Wu et al., 2020). The PTR-ToF-MS
automatically switches between $H_3O^+$ and $NO^+$ modes every 10-20 minutes. In each
mode, the background and ambient measurements were automatically switched to a
custom-built Platinum catalytic converter heated to 365 °C for 3 minutes to detect
background of the instrument. The time resolution of the measurement of PTR-ToF -
MS was 10 s. A total of 31 VOCs species were calibrated using either gas cylinders or
liquid standards. For other measured VOCs, we used the method proposed by Sekimoto
et al. (2017) to determine the relationship between VOC sensitivity and kinetic rate
constants for proton transfer reactions of $H_3O^+$ with VOCs. The fitted line was used to
determine the concentrations of those uncalibrated species. Following the discussions
in Sekimoto, et al. (Sekimoto et al., 2017), the uncertainties of the concentrations for
uncalibrated species were about 50 %. Humidity dependencies of various VOCs were
determined in the laboratory with absolute humidity in the range of 0–30 mmol/mol
(relative humidity of 0 %–92 % at 25 ℃), which fully covered the humidity range
encountered during the entire campaign. The detailed introduction of this method has
been reported by Wu et al. (Wu et al., 2020).
Notably, PTR-ToF-MS is not capable of distinguishing isomers (Yuan et al., 2017).
GC-MS technique was used to measure several carbonyls that PTR-ToF-MS can not
distinguish, including acetaldehyde, propionaldehyde, n-butanal, n-pentanal, n-hexanal,
methacrolein, methyl vinyl ketone. An iodide time-of flight chemical ionization mass
spectrometer (ToF-CIMS) was used to measure propionic acid. Combined with the
measurements of GC-MS and CIMS, the isomers measured by PTR-ToF-MS can be
distinguished. In OVOC species, hydroxyacetone and propionic acid ($C_3H_6O_2$), acetone
and propanal ($C_3H_6O$), methyl ethyl ketone and butanal ($C_4H_8O$), MVK and MACR
($C_4H_6O$) are all isomers. The average concentration of propionic acid measured by
CIMS was 0.23 ppb, significantly lower than that of the concentration of $C_3H_6O_2$



measured by PTR-ToF-MS (~1.5 ppb). The hydroxyacetone concentrations were
determined by the difference between PTR-ToF-MS and CIMS measurements.
Meanwhile, the concentration of propionaldehyde (average of 0.35 ppb) and n-butanal
(average of 0.17 ppb) measured by GC-MS were also respectively far lower than the
concentration of $C_3H_6O$ (average of 4.4 ppb) and $C_4H_8O$ (average of 1.8 ppb) measured
by PTR-ToF -MS. The concentrations of acetone and methyl ethyl ketone were
determined by the difference between PTR-ToF-MS and GC-MS measurements. The
concentrations of MVK and MACR were determined according to $C_4H_6O$
concentration measured by PTR-ToF-MS and the ratio of MVK to MACR measured by
GC-MS. Additionally, the concentration of $CH_4O_2$ and $CH_4O_3$ were also quantified,
which were tentatively attributed to methyl hydroperoxide ($CH_3OOH$) and
hydroxymethyl hydroperoxide ($HOCH_2OOH$), respectively. Additionally, we also
measured concentrations of several small carbon-number acids, including formic acid,
acetic acid, and propionic acid (**Figure S1**). However, the photolysis wavelength bands
of these species are all less than 260 nm. Given the sunlight that can reach the ground
is generally greater than 290 nm, these small carbon-number acids cannot photolyze
significantly near the ground. An exception is pyruvic acid which is also a small carbon-
number acid but with a wide photolysis band that can reach 460 nm because of its
carbonyl functional group (Horowitz et al., 2001;Mellouki and Mu, 2003;Berges and
Warneck, 1992). Therefore, the photolysis of pyruvic acid was included in the analysis
as it can significantly contribute to the production of $RO_X$ radicals.
In addition to the specific species mentioned above, PTR-ToF-MS measured
carbonyls with higher carbon number including $C_nH_{2n}O$ (n>5), $C_nH_{2n-2}O$ (n>3), $C_nH_{2n-2}O_2$ (n>3), $C_nH_{2n-4}O_2$ (n>3) and $C_nH_{2n-4}O_3$ (n>3). Apparently, multiple isomers that
can't be distinguished specifically may contribute to these species. The measured
photodegradable OVOCs species and their concentrations are summarized in **Table S1**.

**2.2 Other measurements**
HONO was measured by a custom-built LOPAP (LOng Path Absorption
Photometer) based on wet chemical sampling and photometric detection (Yu et al.,





2021). HCHO was measured by a custom-built instrument based on the Hantzsch
reaction and absorption photometry. Total OH reactivity was measured by the
comparative reactivity method (CRM) (Sinha et al., 2008;Wang et al., 2021). In this
method, pyrrole ($C_4H_5N$) was used as the reference substance and was quantified by a
quadrupole PTR-MS (Ionicon Analytic, Austria). Non-methane hydrocarbons
(NMHCs) were measured using a gas chromatography-mass spectrometer/flame
ionization detector (GC-MS/FID) system, coupled with a cryogen-free pre-
concentration device. Nitrogen oxides ($NO_X = NO + NO_2$), ozone ($O_3$), sulfur dioxide
($SO_2$) and carbon monoxide (CO) were measured by $NO_X$ analyzer (Thermo
Scientific, Model 42i), $O_3$ analyzer (Thermo Scientific, 150 Model 49i), $SO_2$ analyzer
(Thermo Scientific, Model 43i) and CO analyzer (Thermo Scientific, Model 48i). The
meteorological data, including temperature (T), relative humidity (RH) and wind
speed and direction 160 (WS, WD) were recorded by Vantage Pro2 Weather Station
(Davis Instruments Inc., Vantage Pro2) with the time resolution of 1 min. Photolysis
frequencies including j(HONO), j($NO_2$), j($H_2O_2$), j(HCHO) and j($O^1D$) were
measured by a spectrometer (Focused Photonics Inc., PFS-100).
**2.3 Observation-based box model**

A zero-dimensional box model coupled with the Master Chemical Mechanism

(MCM) v3.3.1 chemical mechanism (Jenkin et al., 2003;Saunders et al., 2003) was used
to simulate $RO_X$ production and losses, and $O_3$ production rates during the field
campaign. The model simulation was constrained to the observations of meteorological
parameters, photolysis frequencies, and concentrations of non-methane hydrocarbons
(NMHCs), OVOCs, NO, $NO_2$, $O_3$, CO, $SO_2$ and nitrous acid (HONO). All constraints
were averaged to generate a synchronized 1-h time resolution dataset. The model runs
were performed in a time-dependent mode with spin-up of two days. A 24-h lifetime
was introduced for all simulated species, including secondary species and radicals, to
approximately simulate dry deposition and other losses of these species (Lu et al.,
2013;Wang et al., 2020b). This lifetime corresponds to an assumed deposition velocity
of 1.2 cm s$^{-1}$ and a well-mixed boundary layer height of about 1 km. Sensitivity tests



show that this assumed deposition lifetime has a relatively small influence on the
reactivity of modeled oxidation products, $RO_X$ radicals and ozone production rates. The
ozone production rate ($P(O_3)$) were calculated according to E1:

$P(O_3) = k_{HO2+NO}[HO_2][NO] + \sum_i(k^i_{RO2+NO}[RO^i_2][NO])$        E1

The production rate of $RO_X$ radicals ($P(RO_X)$) is equal to the sum of the rates at

which all photodegradable species generate $RO_X$ radicals through the photolysis
reactions, as shown in E2.

$P(RO_X) = 2 \times [O_3] \times j(O^1D) \times \theta + [HONO] \times j(HONO) + \sum_i[OVOC_i] \times j_i \times k_i$        E2

where $\theta$ is the fraction of $O^1D$ from ozone photolysis that reacts with water vapor.
$OVOC_i$ represents each OVOCs species, $j_i$ represents the photolysis frequency of each
OVOC species, and $k_i$ represents the number of $RO_X$ radical generated from the
photolysis of each OVOC molecule. For most OVOCs species, $k_i$ is equal to 2.

The photolysis frequencies of measured photodegradable species were calculated

based on measured actinic flux combined with absorption cross sections and
photolysis quantum yields reported in Jet Propulsion Laboratory (JPL) publication
(Burkholder et al., 2020).However, absorption cross sections and photolysis quantum
yields for nitrophenol and methyl nitrophenol are unavailable from JPL publication.
Yuan et al. (2016) have reported that photolysis was the most efficient loss pathway
for nitrophenol in the gas phase. Different values of absorption cross sections and
quantum yields for nitrophenol have been reported (Chen et al., 2011;Sangwan and
Zhu, 2018;Bejan et al., 2006). In this study, we used the values from Chen et al. (Chen
et al., 2011), which can reproduce well the observed concentrations of nitrophenol and
methyl nitrophenol during the measurement period.

Absorption cross sections and quantum yields are not available for carbonyls

with large carbon number, and absorption cross sections and quantum yields of
species with similar structure are used as a surrogate, following the method described
in Jenkin et al., (1997) (Jenkin et al., 1997) (e.g. $C_2H_5C(O)CH_3$ is used as a surrogate
for aliphatic ketones with more carbons). Another issue is that carbonyls with large
carbon number ($C_nH_{2n}O$, n>5; $C_nH_{2n-2}O$, n>3; $C_nH_{2n-2}O_2$, n>3; $C_nH_{2n-4}O_2$, n>3; $C_nH_{2n-4}O_3$, n>3) measured by PTR-ToF-MS may include contributions from multiple


isomers, and the fraction of each individual species cannot be obtained. Hence, each
molecular formula corresponds to multiple molecular structures and thus corresponds
to multiple photolysis frequencies. Here, we calculate the $P(RO_X)$ of these species in
two scenarios: (1) each molecular formula corresponds to minimum photolysis
frequency of all potential species (e.g. aliphatic ketones); (2) each molecular formula
corresponds to maximum photolysis frequency of all potential species (e.g.
aldehydes). As a result, photolysis frequencies of these carbonyls with large carbon
number were assigned to the ranges of $1.2\times10^{-6}\sim6.5\times10^{-6}$, $1.2\times10^{-6}\sim6.5\times10^{-6}$, $1.2\times10^{-6}$
$^{-6}\sim1.2\times10^{-4}$,$1.2\times10^{-6}\sim3.0\times10^{-4}$ and $1.2\times10^{-6}\sim1.8\times10^{-4}$ $s^{-1}$, respectively (Jenkin et al.,
1997) (**Table S1**). The lowest and highest values of these photolysis frequencies were
separately used to determine the lower and upper limits of $P(RO_X)$. Therefore, the
total $P(RO_X)$ contributed by all these OVOC species could be investigated.

## 3 Results and discussion

### 3.1 Overview of the observations

During the observation period, we used PTR-ToF-MS and GC-MS technology to
measure more than 20 photodegradable OVOCs species. The concentrations and
photolysis frequencies of measured photodegradable OVOCs species are summarized
in **Table S1 and Figure 1**. Previous studies have reported that these species have
relatively large absorption cross section and quantum yield (Burkholder et al., 2020).
The measured daytime average photolysis frequencies for these species were generally
larger than $1.3\times10^{-6}$ $s^{-1}$.
**Figure 1** presents the average diurnal variation of photodegradable OVOCs
species during the measurement period. The concentrations of these species ranged
from 0.01 to 10 ppb. HCHO, methylglyoxal, propionaldehyde, n-butanal, n-pentanal,
MVK+MACR, pyruvic acid, formic acid, acetic acid, and $CH_3OOH$ had similar diurnal
variation patterns. The concentrations of these species started to increase from about
6:00 in the morning, and peaked at 13:00-16:00, after which the concentrations
gradually decreased. This diurnal variation pattern is a typical secondary production



pattern, and thus we deduce that these species primarily came from secondary
production. Acetaldehyde, acetone and acrolein showed diurnal variations without
significant variations throughout the day, as these species were contributed by both
secondary generation and primary emissions or background contribution (Wu et al.,
2020). It is notable that acrolein, nitrophenol and methylnitrophenol all peaked at 20:00
in the evening, which is likely due to primary emissions e.g. biomass burning (Ye et al.,

2021).

The ratio of secondary OVOCs to NMHCs can characterize the degree of the
conversion of emitted NMHC to secondary OVOCs through oxidation reactions.
**Figure S2** presents the correlation between daily daytime average of HCHO (and
pyruvic acid) concentration versus OH reactivity from hydrocarbons, i.e.,
HCHO/$k_{OH\_NMHC}$ ratio (and pyruvic acid/$k_{OH\_NMHC}$ ratio) and j(NO$_2$). Both
HCHO/$k_{OH\_NMHC}$ and pyruvic acid/$k_{OH\_NMHC}$ ratios displayed significant positive
correlation with j(NO$_2$). These results suggest that the enhancement of the photolysis
rates converted more NMHCs into secondary OVOCs, suggesting the crucial role of
photolysis reactions in the airmass aging and the occurrence of secondary pollution.
**3.2 Contribution of photolysis reactions to the production of RO$_X$ radicals**
The photolysis of O$_3$, HONO and OVOCs are the most important contributors to
the production of RO$_X$ radicals. All observed photodegradable species, including O$_3$,
HONO and OVOCs, were constrained in the box model to calculate P(RO$_X$). The
calculated P(RO$_X$) was basically determined by concentrations of these observed
photodegradable species. Using the possible ranges of photolysis frequencies of
carbonyls with more carbon number that are not possible to assign into specific
individual species, we can obtain the possible widest variation range of P(RO$_X$). As
shown in **Figure 2a**, the minimum (solid line) and maximum (dashed line) of P(RO$_X$)
calculated during the campaign peaked at 3.6 ppb h$^{-1}$ and 5.4 ppb h$^{-1}$, respectively. The
P(RO$_X$) determined in this study is very close to those reported in the Autumn 2014 in
Pearl River Delta with peak values of 3 ~ 4 ppb h$^{-1}$ (Tan et al., 2019a) and the summer
2014 in Wangdu, Hebei (peak value of 5 ppb h$^{-1}$) (Tan et al., 2017), and lower than



those in the summer 2006 in Beijing (peak value of about 7 ppb h$^{-1}$) (Lu et al., 2013)
and the summer 2006 in Guangzhou (peak value of about 10 ppb h$^{-1}$) (Lu et al., 2012),
and higher than those in the winter of 2016 in Beijing (peak value of about 1 ppb h$^{-1}$)
(Tan et al., 2018) and the winter in the oil and gas basin of Utah, USA (daytime average
value of 0.77 ppb h$^{-1}$) (Edwards et al., 2014). Note that these previous studies mentioned
above usually only measured a few simple carbonyls such as HCHO, acetaldehyde and
acetone and the P(RO$_X$) contributed by photolysis of other OVOCs was calculated by
model simulations, which may lead to large uncertainties.
For the scenario of minimum OVOCs contribution, HONO contributed the most
to P(RO$_X$) (37%), followed by O$_3$ (20%) and HCHO (21%). The contribution of non-
HCHO OVOCs was 22% (**Figure 2a**). For the scenario of maximum OVOCs
contribution, the contribution of non-HCHO OVOCs increased to 44%. In total,
OVOCs contributed 43% ~ 59% of P(RO$_X$), which was higher than the contribution of
HONO. This is different from previous studies reporting HONO contributed more to
P(RO$_X$) than OVOCs in China (Tan et al., 2019a;Tan et al., 2017;Tan et al., 2018;Tan
et al., 2019b). Nevertheless, it is notable that the contributions of HONO to P(RO$_X$) in
the early morning were higher than those of OVOCs due to the accumulation of HONO
in nighttime, while OVOCs dominate P(RO$_X$) at noon when photochemistry was most
active (**Figure 2a**). Furthermore, previous studies in China indicated that HCHO was
the dominant contributor to P(RO$_X$) among OVOC species and the contributions of
other OVOC species was generally smaller than that of HCHO (Tan et al., 2019a;Tan
et al., 2017;Tan et al., 2018;Tan et al., 2019b). In contrast, the results of this study
suggest that non-HCHO OVOCs have a potential to be a larger contributor than HCHO
and HONO, revealing the importance of non-HCHO OVOCs in radical production. The
difference between this study and previous studies in China is primarily attributed to
more OVOC species measured in this study than previous studies. Nevertheless, the
existing isomers of carbonyls with more carbons lead to large uncertainties in the
quantification of P(RO$_X$) as shown in **Figure 2a**. Therefore, precise distinction of these
isomers in the future is crucial to accurately quantify P(RO$_X$). In addition, absorption
cross-section and quantum yield of many photodegradable OVOC species with large


carbon numbers, especially multifunctional species, are not experimentally determined.
As a result, the photolysis frequencies of these species are not available, which also
leads to uncertainties in quantifying P(ROX). As measurements of many organic
compounds may not be possible at least in the near future, construction of
parameterization method for photolysis frequencies of oxygenated VOCs either based
on chemical formula or functional groups at isomeric level will help to reduce this
uncertainty in the future.

As a comparison with the observation-determined P(ROX), P(ROX) was also

simulated by the box model without all observed OVOC species constrained. As shown
in **Figure 3a**, the simulation of the box model without all observed OVOC species
constrained underestimated P(ROX) significantly compared to observation-determined
P(ROX). The underestimation of P(ROX) was 16% and 44% when using the lower and
higher limits of OVOCs photolysis frequencies, respectively (red solid line and red
dashed line in **Figure 3a**). The underestimation of P(ROX) was due to the
underestimation of photodegradable OVOCs simulated by the photochemical model
(**Table S2**). In general, most photodegradable OVOCs were underestimated by 10~100%
by box model except for MVK and MACR. The underestimation of photodegradable
OVOCs can be caused by missing primary emissions (McDonald et al., 2018;Karl et
al., 2018;Gkatzelis et al., 2021) or unknown secondary source of these OVOCs species
(Bloss et al., 2005;Ji et al., 2017). Direct flux measurements of VOCs based on the eddy
covariance technique showed that the contribution of typical urban emission sources
comprised of a surprisingly large portion of OVOCs (Karl et al., 2018). In addition,
some experimental studies indicated that MCM mechanism generally underestimated
formation of aldehydes, ketones and phenols from the oxidation of aromatics by OH
radical (Bloss et al., 2005;Ji et al., 2017), suggesting the existence of unknown
secondary source of these OVOCs species. This evidence suggests that it is essential to
use ambient measurements of OVOCs as constraints in models at least until primary
and secondary sources of OVOCs are better understood.

Previous studies in Pearl River Delta and North China Plain of China found that

photochemical models significantly underestimated measured concentrations of OH





radicals, indicating the existence of unknown sources of $RO_X$ radicals in the atmosphere
(Lu et al., 2012;Lu et al., 2013;Tan et al., 2019c;Hofzumahaus et al., 2009;Ma et al.,
2019). For instance, comprehensive measurements in winter in Beijing showed that the
photochemical box model greatly underestimated OH, $HO_2$ and $RO_2$ radicals by 50%
~ 12 fold during the pollution periods (Tan et al., 2018;Ma et al., 2019). Through the
budget analysis of the source and sink of radicals, the researchers believed that the
missing $P(RO_X)$ was the primary cause of the underestimation of $HO_2$ and $RO_2$
concentrations (Tan et al., 2018). Given that most photodegradable OVOCs were not
constrained in box model used in these previous studies of $RO_X$ radicals, the results of
our study provide a direction for solving this issue regarding underestimated ROx
radical concentrations. Therefore, it is imperative to continuously improve
measurement technologies to achieve accurate quantification of more photodegradable
OVOC species, thereby improving our understanding of the issues with respect to the
closure of $RO_X$ radicals in the atmosphere.
**3.3 The role of photolysis reactions in ozone pollution**
The box model was used to evaluate the effect of the photodegradable OVOCs
species on ozone production rate during the whole campaign. $P(O_3)$ were simulated
with and without all of measured photodegradable OVOCs species constrained in the
box model, respectively. As shown in **Figure 3b**, compared to the scenario with
observed photodegradable OVOCs species constrained in box model, the scenario
without constraining OVOCs underestimated peak value of $P(O_3)$ by 15~38%. The
underestimation of $P(O_3)$ was due to the underestimation of OVOCs by the box model
(**Table S2**). As shown in **Figure 4,** the dependence of daily peak $O_3$ concentrations on
$NO_X$ concentrations was calculated by the box model with and without all of
measured photodegradable OVOCs species constrained. The $NO_X$ concentration level
corresponding to maximum of ozone concentration ($NOx\ (O_{3\ max})$) was determined. In
reality, this $NO_X$ concentration level is the threshold to distinguish between VOC-
limited and $NO_X$-limited regimes (Edwards et al., 2014;Womack et al., 2019). Ozone
production is $NO_X$-limited if the ambient $NO_X$ concentration is lower than the





368 threshold of NO$_X$, but is in VOC-limited regime if ambient NO$_X$ concentration higher

369 than the threshold of NO$_X$. The larger threshold of NO$_X$ represents higher possibility

370 of ozone production in NO$_X$ limited regime. The threshold of NO$_X$ for the scenario

371 with observed photodegradable OVOCs species constrained is 21%~52% higher than

372 that without observed photodegradable OVOCs species constrained (**Figure 4)**. This

373 suggests that the box model simulation without constraining OVOCs will

374 overestimate the VOC-limited degree due to the underestimation of OVOCs, and thus

375 overestimate the effect of VOCs reduction in reducing ozone pollution, which in turn

376 may not determine the ozone control strategy correctly. Therefore, it is necessary to

377 constrain these important photodegradable species in photochemical models to

378 calculate P(O$_3$) level and to diagnose ozone sensitivity regimes accurately.

379  O$_3$ production rate can be expressed as the product of P(RO$_X$) and radical chain

380 length (ChL) as shown in E3 (Tonnesen and Dennis, 2000).

381 $$P(O_3) = P(RO_X) \times \frac{Rate(HO_2+NO)+Rate(RO_2+NO)}{P(RO_X)} = P(RO_X) \times ChL \qquad \text{E3}$$

382  where Rate (HO$_2$+NO) and Rate (RO$_2$+NO) represent the reaction rates between

383 HO$_2$ and NO and between RO$_2$ and NO, respectively.

384  Two ozone pollution episodes (from 19 September to 27 September and from 30

385 September to 9 October, respectively) were identified during the campaign from 14

386 September to 20 November 2018 (**Figure S3, Table S3**). The temporal variations of

387 P(O$_3$) and P(RO$_X$) overall showed good consistency with those of ozone concentrations

388 (**Figure S4**). P(O$_3$) and P(RO$_X$) in the two ozone pollution episodes are higher than

389 those in the non-pollution period (**Figure 5, Figure S5**). ChL levels were similar

390 between the ozone pollution episodes and the non-pollution period (**Figure S5**).

391 Therefore, the substantial increase of P(RO$_X$) in the ozone pollution episode played a

392 crucial role in the accelerated ozone production. Furthermore, the ratio of P(RO$_X$) from

393 OVOCs photolysis to total P(RO$_X$) in the two ozone pollution episodes is higher than

394 that in the non-pollution period, denoting higher contribution of OVOCs photolysis to

395 P(RO$_X$) in the ozone pollution episodes (**Figure 5**). These results indicate that the

396 accelerating production of OVOCs had a significant positive feedback effect on ozone





pollution. This is broadly consistent with the wintertime observations in an oil and gas
basin in Utah, USA, which found that a very high VOC to $NO_X$ ratio optimized
production of secondary OVOCs, leading to OVOC photolysis as a dominant oxidant
source (Edwards et al., 2014).

**4 Summary and Conclusion**


In summary, comprehensive measurements of photodegradable species advance
our understand of radical sources and ozone production in an urban environment. By
using PTR-ToF-MS in a representative urban environment, a large number of
photodegradable OVOCs were measured. These measurements make it possible to
directly quantify their contribution to $RO_X$ radical production. We found that non-
HCHO OVOCs can be a larger contributor to $P(RO_X)$ than HCHO and HONO.
Photochemical models without constrained OVOC species will significantly
underestimate $P(RO_X)$ and ozone production rates and overestimate the effect of VOCs
reduction in reducing ozone pollution. Therefore, it is important to measure these
photodegradable species and use these observations as constraints to better quantify
radical and ozone production.
Thanks to the improvement of technology in the recent years, large number of
OVOCs species in the atmosphere can be measured by the emerging online chemical
ionization mass spectrometers, including PTR-ToF-MS and CIMS. However,
photolysis frequencies of these OVOCs species, especially those with multiple
functional groups, are still not available or difficult to quantify using current existing
information, which poses large uncertainties in the quantification of $P(RO_X)$ and ozone
production. Hence, more laboratory studies on photolysis of organic compounds, better
parameterization of photolysis frequencies using chemical formula/functional groups,
and measurements of oxygenated VOCs at isomeric level will help to decrease this
uncertainty in the future.



**Data availability**


The observational data used in this study are available from corresponding authors
upon request (byuan@jnu.edu.cn)

**Author contributions**


BY, WJW and HS designed the research. WJW and BY prepared the manuscript
with contributions from other authors. WJW performed data analysis with contributions
from YWP, YFC, SXY and FXB. CHW, JPQ, YBH, CMW, CSY, ZLW, BLW, XMW,
WS, WWH, PC, MNZ, JYZ, and MS collected data

**Competing interests**


The authors declare that they have no known competing financial interests or personal
relationships that could have appeared to influence the work reported in this paper.

**Acknowledgements**


This work was supported by the National Key R&D Plan of China (grant No.
2019YFE0106300), the National Natural Science Foundation of China (grant No.
41877302, 41905111), Guangdong Natural Science Funds for Distinguished Young
Scholar (grant No. 2018B030306037), Key-Area Research and Development Program
of Guangdong Province (grant No. 2019B110206001), Guangdong Soft Science
Research Program (grant No. 2019B101001005), and Guangdong Innovative and
Entrepreneurial Research Team Program (grant No. 2016ZT06N263). This work was
also supported by Special Fund Project for Science and Technology Innovation Strategy
of Guangdong Province (Grant No.2019B121205004).





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



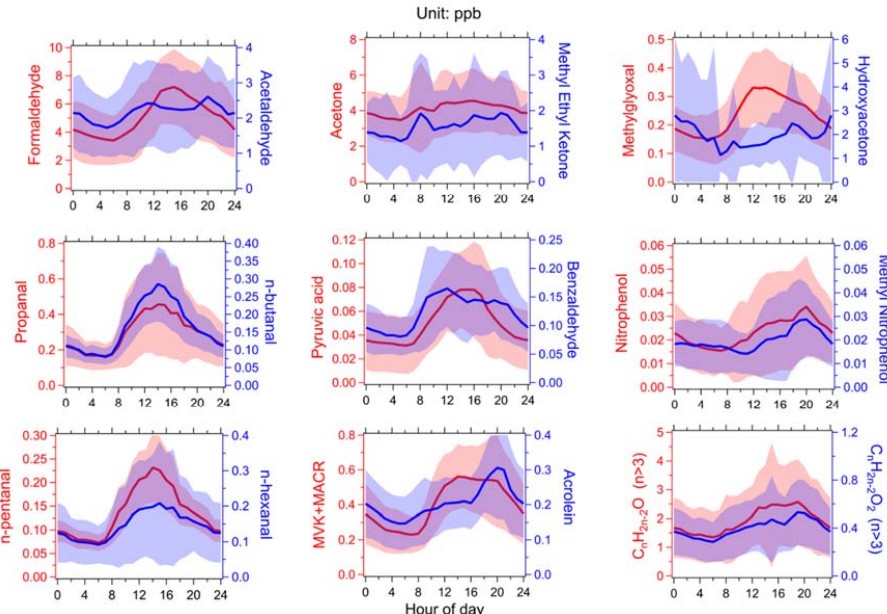

Figure 1. The average diurnal variations of the concentrations of photodegradable OVOCs species during the field campaign in Guangzhou.

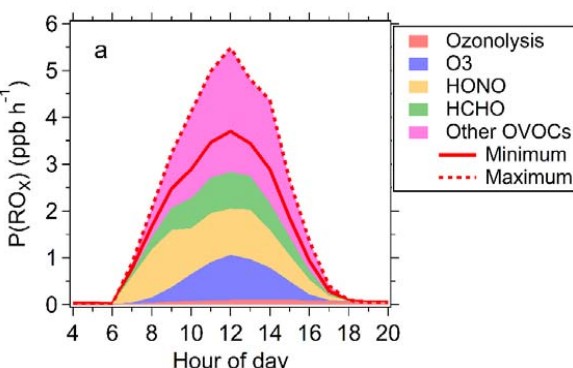

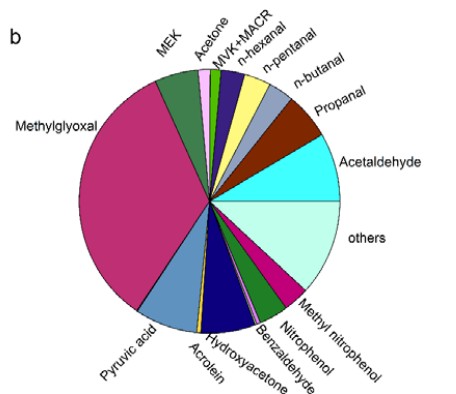



Figure 2. The P(RO$_X$) calculated by box model with all observed photodegradable
species constrained. (a): The source composition of total P(RO$_X$) during the campaign;
the solid and dashed lines represent the scenarios with minimum and maximum OVOC
contributions to P(RO$_X$), respectively. (b): the relative contributions of non-HCHO
OVOC species to P(RO$_X$) for the scenarios with minimum OVOC contribution to
P(RO$_X$).



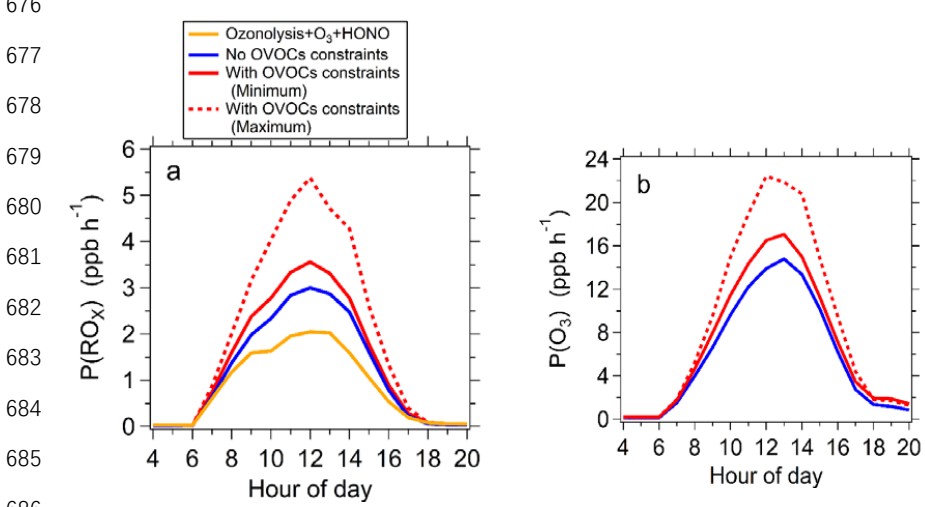

Figure 3. Model simulated P(RO$_X$) (a) and P(O$_3$) (b) without and with all observed photodegradable OVOCs constrained. (a): Model simulated P(RO$_X$) without (blue line) and with all observed photodegradable OVOCs constrained (red lines). The sum contribution of O$_3$ photolysis, HONO photolysis and ozonolysis is also displayed (yellow line). (b): Model simulated P(O$_3$) without (blue line) and with observed photodegradable OVOCs constrained (red lines). The red solid and red dashed lines represent the scenarios with minimum and maximum OVOC contributions to P(RO$_X$), respectively.





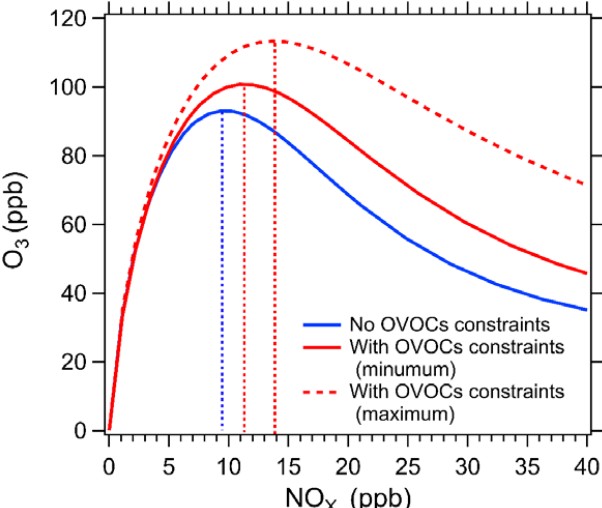

698

Figure 4. Model simulated dependence of daily peak $O_3$ concentrations on $NO_X$ concentrations without (blue curve) and with all observed photodegradable OVOCs constrained (red curves). The red solid and red dashed curves represent the scenarios with minimum and maximum OVOC contributions to $P(RO_X)$, respectively. The dashed lines parallel to Y-axis represent the threshold of $NO_X$ levels to distinguish between VOC-limited and $NO_X$-limited regimes.





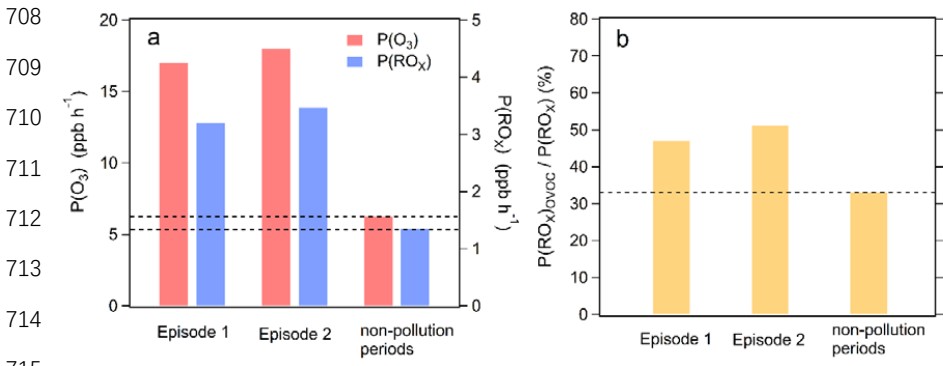

Figure 5. Averaged P(O$_3$), P(RO$_X$), the ratio of P(RO$_X$) contributed by OVOCs to total

P(RO$_X$) (P(RO$_X$)$_{OVOC}$/P(RO$_X$)) during two ozone pollution episodes (episode 1,

episode 2) and non-pollution periods. Both P(O$_3$) and P(RO$_X$) correspond to the

scenarios with minimum OVOC contributions to P(RO$_X$).