# Peer review of "Direct observations indicate photodegradable oxygenated VOCs"

_Atmospheric Chemistry and Physics, 2021_

## Referee Comment (RC2)

Review of "Direct observations indicate photodegradable oxygenated VOCs as larger contributors to radicals and ozone production in the atmosphere," Wang et al., ACP (2021)

**Summary**

This paper analyzes ground-based observations of traces gases acquired in an urban area of China during Fall 2018. The analysis focus is on oVOC observations and the contribution of oVOC to radical and ozone production. Using a box model, the study demonstrates that oVOC photolysis contributes significantly to primary radical production. The implication is that models that do not capture oVOC mixing ratios correctly (either due to missing emissions, inadequate chemical production, or other reasons) will have errors in calculation of ozone production rates and regimes. Publication is recommended following minor revisions.

**Specific Comments**

L118: Does this uncertainty influence the calculation of PROx from oVOC? If so, how much? I assume it is proportional to the PROx contributions of these unmeasured compounds. A simple way to calculate the error budget for this might be to take a root-sum-square of the uncertainties weighted by their PROx contributions for each species. This might provide some insight into which species we need to focus on for improved measurements/calibration. Just a suggestion.

L190: It is unnecessary to equate this to a deposition velocity. The 24h lifetime is a standard method in 0-D box modeling to account for all types of "physical loss," which might also include entrainment, advection, and other things. At least that's the hand-waving justification; the truth is that it is required to limit buildup of secondary species, because a 0-D model is an approximation of a 3-D atmosphere.

L191: Over what range were the sensitivity tests conducted?

L192: change "deposition" to "physical loss".

L255: Is this due to home heating, or from wild/agricultural fires?

L259: Multiplying the y-axis values in Fig. S2 by k(OH+HCHO) or k(OH + pyruvic) might improve interpretation. Then, the scale is a unitless fractional contribution to OH reactivity. Also, it should be mentioned here and in the figure captions that each point is an average over one day of the campaign (I assume).

L265: You mention the role of secondary pollution. Do the y-axis intercepts in Fig. S2 have any significance for contributions of primary emissions?

L292 – 300: It would be more pertinent here to provide some quantitative metrics for how important oVOC was as a HOx source in these previous studies (either in ppb/h or %) to compare to the present analysis.

L348: How much does OH and HO2 increase between the case with oVOC constrained or not? It seems unlikely this would be a difference of 50% or more. Unless the oVOC contributions were even larger in those previous studies featuring HOx measurements?

L379: It would help less experienced readers to have some discussion/description on the conceptual definition of chain length.

L389: ChL for episode 1 is double the non-pollution value at mid-day (by eyeball).

L396: This notion of radical amplification is also discussed in Qu et al. (2021), and that should probably be referenced here.

L667: Is glyoxal included in the "others" block? Or is it not included here because it was not measured? I would expect glyoxal to be comparable to methylglyoxal as a ROx source. If this figure is limited to observed oVOC, this should be stated clearly in the caption and probably discussed in the text, since there are likely even more oVOC than those observed by the PTRMS.

L687: Does the blue line use the "minimum" oVOCs contribution? If so, I think there should also be a dashed blue line for the maximum case, or just eliminate the dashed red line. As it stands, it is not clear what the blue line should be compared to.

**Technical Comments**

L34: "comparable to"

L49: replace "hydroxide" with "hydrogen oxide"

L51: Recommend replacing "Photodegradable species…including" with "Photolysis of" and deleting "via photolysis reactions."

L139: PTR-ToF-MS

L202: "radicals"

L207: space after period

L270: delete "basically"

L271: "species and calculated photolysis frequencies derived from observed actinic flux."

L361: Start a new paragraph.

L424: Availability of model code?

L543: Qu reference missing information.

L661: What is shading in Fig. 1?

L667: ozonolysis is in this figure but never mentioned in the paper

Supplement, L57: Define ChL here.

Table S1: 4th column is a mixing ratio, not concentration. Also you may wish to include measurement uncertainty here (your choice).

---

## Author Comment (AC1)

**Responses to Reviewer #1**

Major comments:

How did you calibrate the concentrations of acetaldehyde, propionaldehyde, n-butanal, n-pentanal, n-hexanal, methacrolein, and methyl vinyl ketone measured by the GC-MS technique? These OVOCs may be unstable in cylinders. Additionally, the preconcentration procedure for the GC-MS technique may cause loss of the OVOCs especially under high RH condition because of their relatively high Henry constant with respect to NMHCs. The authors are suggested to present the detail information about the calibration in the section of Material and Methods.

Reply: Many thanks for your suggestion. Cylinder standard gases were employed to calibrate the online GC-MS/FID system. A 63-chemicals mixture standard (Spectra Gases) was used to calibrate C2–C6 carbonyls, methanol, and halocarbons. Concentrations of acetaldehyde, acetone, propionaldehyde, n-butanal, n-pentanal, n-hexanal, methacrolein, and methyl vinyl ketone were calculated according to their calibration curves. We agree with you that the GC-MS technique may have some uncertainty in measuring OVOCs. Therefore, we compared OVOC concentrations measured by GC-MS and PTR-ToF-MS. The measurement results of the two instruments are quite similar. Therefore, we think the uncertainty of the GC-MS technique is acceptable. This gives us confidence to use GC-MS data. I added the detailed calibration procedure of GC-MS in Supplement.

**Line 131-133: We compared concentrations of common OVOC species measured by both GC-MS and PTR-ToF-MS. The agreement of measurement results from the two instruments are quite consistent (Figure S1).**

**Line 25-38 in Supplement: Cylinder standard gases were employed to calibrate the online GC-MS/FID system. A 63-chemicals mixture standard (Spectra Gases) was used to calibrate C2–C6 OVOCs and halocarbons. The calibration curves**

for each species were acquired by diluting the mixture standard gas into five concentration gradients. The coefficients of determination ($r^2$) for all calibration curves are larger than 0.995. Ambient concentrations of C2–C6 OVOCs were calculated according to their calibration curves. It is worth noting that these OVOCs may be unstable in cylinders. Additionally, the preconcentration procedures for the GC-MS technique may cause loss of the OVOCs especially under high RH condition because of their relatively high Henry constants. We compared OVOC concentrations measured by GC-MS and PTR-ToF-MS. The measurement results of the two instruments are quite similar (Figure S1). Therefore, the uncertainty of the GC-MS technique is acceptable. This gives us confidence to use OVOC data from GC-MS.

[Figure]

Figure S1. Comparison of typical OVOC concentrations measured by both GC-MS and PTR-ToF-MS.

Both propionaldehyde and acetone have signals in the GC-MS. Why did you derive acetone concentration from the difference between PTR-ToF-MS and GC-MS measurements?

Reply: Many thanks for your suggestion. Given the higher accuracy of PTR-ToF-MS technology in terms of measuring OVOCs, we gave priority to utilizing the OVOCs data of PTR-ToF-MS for analysis. The difference between PTR-ToF-MS and GC-MS at least helps to reduce the uncertainty of PTR-ToF-MS induced by isomers.

**Line 149-150: In this way, the uncertainty of PTR-ToF-MS induced by isomers is greatly reduced.**

**Line 39-46 in Supplement: Given the higher accuracy of PTR-ToF-MS technology in terms of measuring OVOCs, we gave priority to utilizing the OVOCs data of PTR-ToF-MS for analysis in this study. To reduce the uncertainty of PTR-ToF-MS induced by isomers, the concentrations of acetone were determined by the difference between the $C_3H_6O$ concentrations measured by PTR-ToF-MS and propanal concentrations measured by GC-MS; the concentrations of MVK and MACR were determined according to $C_4H_6O$ concentration measured by PTR-ToF-MS and the ratio of MVK to MACR measured by GC-MS.**

Why did you obtain the concentrations of MVK and MACR by using the C4H6O concentration measured by PTR-ToF-MS and the ratio of MVK to MACR measured by GC-MS?

Reply: Many thanks for your suggestion. Given the higher accuracy of PTR-ToF-MS technology in terms of measuring OVOCs, we gave priority to utilizing the OVOCs data of PTR-ToF-MS for analysis. The ratio of MVK to MACR measured by GC-MS helps to distinguish between MVK and MACR in C4H6O isomers measured by PTR-ToF-MS.

**Line 149-150: In this way, the uncertainty of PTR-ToF-MS induced by isomers is greatly reduced.**

**Line 39-46 in Supplement: Given the higher accuracy of PTR-ToF-MS technology in terms of measuring OVOCs, we gave priority to utilizing the OVOCs data of PTR-ToF-MS for analysis in this study. To reduce the uncertainty of PTR-ToF-MS induced by isomers, the concentrations of acetone were determined by the difference between the $C_3H_6O$ concentrations measured by PTR-ToF-MS and propanal concentrations measured by GC-MS; the concentrations of MVK and MACR were determined according to $C_4H_6O$ concentration measured by PTR-ToF-MS and the ratio of MVK to MACR measured by GC-MS.**

The lifetimes of several species, e.g., isoprene and HONO are usually less than 15 min in noontime, the constrains with 1-h time resolution dataset may significantly underestimate their role in radicals' formation. The authors are suggested to present a brief discussion about the weakness of model simulation.

Reply: Many thanks for your suggestion. I agree with the reviewer about this concern. The relatively low time resolution used in model simulation is determined by the 1-h time resolution of VOC data. Model simulation with 5-min time resolution could be achieved by interpolating VOC data. We compared the OH and $HO_2$ concentrations simulated by the box model with 5-min and 1-hour time resolution. The difference in simulated radical concentrations between 1-h and 5-min time resolution is within 10% (Figure S3). This indicates the 1-h time resolution used in the box model is acceptable.

**Lines 197-199: There is no significant difference in simulated OH and $HO_2$ concentrations between 1-hour time resolution and 5-minute time resolution (Figure S3).**

[Figure]

Figure S3. The simulated radical concentrations with 1-h and 5-min time resolution on October 1, 2018.

The photolysis of OVOCs usually have multichannel with different contribution to ROx radicals, e.g., photolysis of HCHO can produce H2 and CO in one channel or H and HCO in another channel, the former channel makes no contribution to ROx radicals, whereas the later channel contributes to 2 molecules of ROx. Therefore, using the total photolysis frequency of each OVOC with ki value of 2 (in equation E2) must largely overestimate its contribution to ROx radicals, especially for the carbonyls with large carbon numbers measured by PTR-ToF-MS because their photolysis mechanisms are not included into the MCM (v3.3.1).

Reply: Many thanks for your suggestion. We used the photolysis frequencies corresponding to radical formation channel, not including molecule formation channel, such as H2 and CO formed from HCHO photolysis. For common OVOC species such as HCHO, CH3CHO and acetone, the photolysis frequencies were calculated based on measured actinic flux combined with absorption cross sections and photolysis quantum yields reported in Jet Propulsion Laboratory (JPL) publication. For carbonyls with large carbon numbers measured by PTR-ToF-MS, absorption cross sections and quantum yields are not available, and absorption cross sections and quantum yields of species with similar structure are used as a surrogate, following the method described in Jenkin et al., (Jenkin et al., 1997) (e.g. $C_2H_5C(O)CH_3$ is used as a surrogate for aliphatic ketones with more carbons). Note that all of absorption cross sections and quantum yields used here corresponds to

radical formation channel, not including molecule formation channel. ki value in equation E2 for most species is 2, but is 1 for some species. We have added this discussion in the manuscript.

**Line 218-221: The photolysis frequencies of measured photodegradable species were calculated based on measured actinic flux combined with absorption cross sections and photolysis quantum yields reported in Jet Propulsion Laboratory (JPL) publication (Burkholder et al., 2020).**

**Line 221-223: Note that absorption cross sections and quantum yields used all correspond to radical formation channel, not including molecule formation channel.**

**Line 231-235: Absorption cross sections and quantum yields are not available for carbonyls with large carbon number, and absorption cross sections and quantum yields of species with similar structure are used as a surrogate, following the method described in Jenkin et al., (Jenkin et al., 1997) (e.g. $C_2H_5C(O)CH_3$ is used as a surrogate for aliphatic ketones with more carbons).**

**Line 217: For most OVOCs species, $k_i$ is equal to 2 or 1.**

**Minor comments:**

The title is suggested to be "Unexpectedly large contribution of oxygenated VOCs to atmospheric radicals and ozone production in Guangzhou".

Reply: Many thanks for your suggestions. After careful consideration, we think that it is no need to emphasize "Guangzhou" in the title because this will limit the application of the study. In addition, we hope to highlight "the direct observations". A recent study by Qu et al. (2021) also reported the large contribution of oxygenated VOCs to atmospheric radicals and ozone production. and this study is based on model

simulation. The strength of our study is that it is based on direct observations rather than solely from model simulations.

Reference: Qu, H., Wang, Y., et al: Chemical Production of Oxygenated Volatile Organic Compounds Strongly Enhances Boundary-Layer Oxidation Chemistry and Ozone Production, Environmental Science & Technology, 55, 13718-13727, 2021.

Lines 30-32, this sentence is suggested to be "a large number of oxygenated VOCs have been quantified in Guangzhou city, China.".

Reply: Thanks. We have revised it.

Lines 32-34, the sentence is suggested to be moved after the sentence in lines 34-37. "contribute" should be "contribute to". "comparable or larger than the contributions from nitrous acid and formaldehyde" is better rephrased as "which is comparable to or larger than the contributions from nitrous acid and formaldehyde".

Reply: Thanks. We have revised it.

Line 39, "will underestimate P(ROX) and ozone production rate" is better to be "will underestimate the production rates of ROX and ozone".

Reply: Thanks. We have revised it.

Line 127, the abbreviations of MACR and MVK for methacrolein and methyl vinyl ketone are suggested to be noted in brackets, or readers may not understand the meanings of the abbreviations appeared in the following.#

Reply: Thanks. We have revised it.

**References:**

Burkholder, J., Sander, S., Abbatt, J., Barker, J., Cappa, C., Crounse, J., Dibble, T., Huie, R., Kolb, C., and Kurylo, M.: Chemical kinetics and photochemical data for use in atmospheric studies; evaluation number 19, Pasadena, CA: Jet Propulsion Laboratory, National Aeronautics and Space …, 2020.

Jenkin, M. E., Saunders, S. M., and Pilling, M. J.: The tropospheric degradation of volatile organic compounds: a protocol for mechanism development, Atmos. Environ., 31, 81–104, 1997.

---

## Author Comment (AC2)

**Responses to Reviewer #2**

Summary

This paper analyzes ground-based observations of traces gases acquired in an urban area of China during Fall 2018. The analysis focus is on oVOC observations and the contribution of oVOC to radical and ozone production. Using a box model, the study demonstrates that oVOC photolysis contributes significantly to primary radical production. The implication is that models that do not capture oVOC mixing ratios correctly (either due to missing emissions, inadequate chemical production, or other reasons) will have errors in calculation of ozone production rates and regimes. Publication is recommended following minor revisions.

Specific Comments

L118: Does this uncertainty influence the calculation of PROx from oVOC? If so, how much? I assume it is proportional to the PROx contributions of these unmeasured compounds. A simple way to calculate the error budget for this might be to take a root-sum-square of the uncertainties weighted by their PROx contributions for each species. This might provide some insight into which species we need to focus on for improved measurements/calibration. Just a suggestion.

Reply: Many thanks for your suggestions. We have included the discussion in the manuscript. We agree with the reviewer that the uncertainty of P(ROx) should be proportional to the P(ROx) contributions of these unmeasured compounds. By taking a root-sum-square of the uncertainties weighted by their P(RO$_X$) contributions for each species, we found that the uncertainty of the concentrations of these species lead to uncertainty of 0.23 ppb h$^{-1}$ in calculation of P(RO$_X$), for the scenario of maximum OVOCs contribution to P(RO$_X$). Given that the photolysis frequencies of carbonyls with large carbon numbers can vary several orders of magnitude, the uncertainty in photolysis frequencies is a more important factor that influence the calculation of P(RO$_X$) from OVOCs than the uncertainty from measurement (only ~50%). To keep the smooth logic of the manuscript, we added this discussion about measurement

uncertainty into supplement.

**Line 47-59 in Supplement: The common OVOCs species were calibrated in this study. However, some OVOC species, including pyruvic acid, nitrophenol, methyl nitrophenol and carbonyls with large carbon number, were not calibrated. For these OVOC species, we used the method proposed by Sekimoto et al. (2017) to determine the relationship between VOC sensitivity and kinetic rate constants for proton transfer reactions of $H_3O^+$ with VOCs. The fitted line was used to determine the concentrations of those uncalibrated species. Following the discussions in Sekimoto, et al. (Sekimoto et al., 2017), the uncertainties of the concentrations for uncalibrated species were about 50 %. The uncertainties in the concentrations of these species lead to uncertainties of 0.04~0.23 ppb h-1 (1.3%~8.0%) in calculation of $P(RO_X)$. Among these species, $C_nH_{2n-2}O_2$ (n>3) contributes the largest uncertainty, followed by $C_nH_{2n-4}O_2$ (n>3), $C_nH_{2n-4}O_3$ (n>3), $C_nH_{2n-2}O$ (n>3), $C_nH_{2n}O$ (n>5), pyruvic acid, nitrophenol and methyl nitrophenol.**

L190: It is unnecessary to equate this to a deposition velocity. The 24h lifetime is a standard method in 0-D box modeling to account for all types of "physical loss," which might also include entrainment, advection, and other things. At least that's the hand-waving justification; the truth is that it is required to limit buildup of secondary species, because a 0-D model is an approximation of a 3-D atmosphere.

Reply: Thank you. We have deleted this sentence.

L191: Over what range were the sensitivity tests conducted?

Reply: We changed physical loss lifetime by 50% to test its effect on OH and $HO_2$ concentrations. A 50% change in physical loss lifetime leads to 3%, 6% and 10% changes in OH concentration, $HO_2$ concentration and $P(O_3)$.

**Line 205-207: A 50% change in the physical loss lifetime leads to 3%, 6% and 10%**

**changes in OH concentration, HO$_2$ concentration and ozone production rate.**

L192: change "deposition" to "physical loss".

Reply: Thank you. We have revised it.

**Line 203-205: Sensitivity tests show that this assumed physical loss lifetime has a relatively small influence on the reactivity of modeled oxidation products, RO$_X$ radicals and ozone production rates.**

L255: Is this due to home heating, or from wild/agricultural fires?

Reply: We think it is more likely from wild/agricultural fires. Home heating is less likely as it was still warm in October and November in Guangzhou region. In addition, vehicle emission is another possible reason because the measurement site is close to heavy traffic.

**Line 270-272: It is notable that acrolein, nitrophenol and methylnitrophenol all peaked at 20:00 in the evening, which is likely due to primary emissions e.g. biomass burning due to wild/agricultural fires (Ye et al., 2021) and vehicle emissions.**

L259: Multiplying the y-axis values in Fig. S2 by k(OH+HCHO) or k(OH + pyruvic) might improve interpretation. Then, the scale is a unitless fractional contribution to OH reactivity. Also, it should be mentioned here and in the figure captions that each point is an average over one day of the campaign (I assume)

Reply: Thank you. We have revised it according to your suggestions. To distinguish the OH reactivity from reaction rate constant, we used R$_{OH}$ rather than k$_{OH}$ in the revised manuscript.

**Line 82 in Supplement:**

[Figure]

Figure S4. The scatter plot of daily daytime average $R_{OH\_HCHO}$ / $R_{OH\_NMHC}$ (and $R_{OH\_pyruvic\ acid}$ / $R_{OH\_NMHC}$) ratios versus j(NO2) color-coded using ozone concentrations during the campaign. Each point corresponds to a daytime average over one day of the campaign. Red lines are the linear fitting of the scatters.

L265: You mention the role of secondary pollution. Do the y-axis intercepts in Fig. S2 have any significance for contributions of primary emissions?

Reply: Yes, I think the y-axis intercepts can characterize contributions of primary emissions to some extent, because there is no secondary production at photolysis frequency value of zero. The intercepts for both $R_{OH\_HCHO}$ / $R_{OH\_NMHC}$ and $R_{OH\ pyruvic\ acid}$ / $R_{OH\_NMHC}$ are close to zero, indicating that primary emissions played a minor role in the concentration of two OVOCs.

L292 – 300: It would be more pertinent here to provide some quantitative metrics for how important oVOC was as a HOx source in these previous studies (either in ppb/h or %) to compare to the present analysis.

Reply: Many thanks. We have added this discussion in the manuscript.

**Line 312-316: In total, OVOCs contributed 43% ~ 59% of P(ROX), which is higher than previous studies that reported OVOCs contributed 17%~40% of P(RO_X) in major cities in China and the US (Tan et al., 2019a;Tan et al., 2017;Tan et al., 2018;Tan et al., 2019b;Young et al., 2012;Griffith et al., 2016).**

L348: How much does OH and HO2 increase between the case with oVOC constrained or not? It seems unlikely this would be a difference of 50% or more. Unless the oVOC contributions were even larger in those previous studies featuring HOx measurements?

Reply: Many thanks. We have added this discussion in the manuscript. The simulation of the box model without all observed OVOC species constrained underestimated $P(RO_X)$ by 16%~44% and underestimated OH and $HO_2$ by 15~38% and 25%~64% respectively, compared to the case with all observed OVOC species constrained.

**Line 347-350: The underestimation of $P(RO_X)$ was 16% and 44% when using the lower and higher limits of OVOCs photolysis frequencies, respectively (red solid line and red dashed line in Figure 3a). In this case, the underestimation of OH and HO2 concentrations were 15~38% and 25%~64%, respectively.**

L379: It would help less experienced readers to have some discussion/description on the conceptual definition of chain length.

Reply: Many thanks. ChL characterizes the number of iterations each $RO_X$ radical makes prior to termination. It is equal to the ratio between the radical recycling rate and primary production rate, indicating the efficiency of radical propagation.

**Line 413-416: ChL characterizes the number of iterations each $RO_X$ radical makes prior to termination. It is equal to the ratio between the radical recycling rate and primary production rate (or equivalently, termination rate), indicating the efficiency of radical propagation.**

L389: ChL for episode 1 is double the non-pollution value at mid-day (by eyeball).

Reply: Many thanks. Yes, ChL for episode 1 was a factor of 1.7 that in non-pollution period. We have modified this part.

**Line 422-426: $P(O_3)$ in the two ozone pollution episodes was a factor of 2.6~2.8 that in the non-pollution period (Figure 5, Figure S8). $P(RO_X)$ in the two ozone**

**pollution episodes was a factor of 2.2~2.6 that in the non-pollution period. ChL in episode 2 was similar to that in non-pollution period, while ChL for episode 1 was a factor of 1.7 that in non-pollution period (Figure S8).**

L396: This notion of radical amplification is also discussed in Qu et al. (2021), and that should probably be referenced here.

Reply: Many thanks. We have included this citation.

**Line 433-434: These results indicate that the accelerating production of OVOCs had a significant positive feedback effect on ozone pollution (Qu et al., 2021).**

L667: Is glyoxal included in the "others" block? Or is it not included here because it was not measured? I would expect glyoxal to be comparable to methylglyoxal as a ROx source. If this figure is limited to observed oVOC, this should be stated clearly in the caption and probably discussed in the text, since there are likely even more oVOC than those observed by the PTRMS.

Reply: Many thanks. This figure corresponds to the simulation result of box model with all observed OVOC species constrained. Therefore, the total $P(RO_X)$ contains all observed OVOCs and simulated OVOCs that were not measured. The simulated OVOCs that were not measured was integrated into the "others" in Figure 2b, which includes glyoxal. Although the concentration of glyoxal (0.37 ppb) was slightly higher than that of methylglyoxal (0.32 ppb), the photolysis frequency of glyoxal ($3.5*10^{-6}$ s$^{-1}$, daytime average) is far lower than that of methylglyoxal ($5.5*10^{-5}$ s$^{-1}$, daytime average). As a result, $P(RO_X)$ from glyoxal (0.010 ppb h$^{-1}$) is far lower than $P(RO_X)$ from methylglyoxal (0.13 ppb h$^{-1}$). Note that Figure 2b shows the relative contributions of different non-HCHO OVOC species to $P(RO_X)$ for the scenarios with minimum OVOC contribution to $P(RO_X)$. We have added the scenario scenarios with maximum OVOC contribution to $P(RO_X)$ in Supplement (Figure S5) of the revised manuscript.

**Line 286-288: The simulated total $P(RO_X)$ contains the contributions from all observed photodegradable species and several simulated OVOCs that was not**

**measured such as glyoxal.**

**Line 88 in Supplement:**

[Figure]

Figure S5. The relative contributions of non-HCHO OVOC species to P(RO$_X$) for the scenarios with maximum OVOC contribution to P(RO$_X$).

**Lines 308-310: Figure 2b and Figure S5 show the relative contributions of different non-HCHO OVOC species to P(RO$_X$) for the scenarios with minimum and maximum OVOC contribution, respectively.**

L687: Does the blue line use the "minimum" oVOCs contribution? If so, I think there should also be a dashed blue line for the maximum case, or just eliminate the dashed red line. As it stands, it is not clear what the blue line should be compared to.

Reply: The blue line represents the scenario without observed OVOCs constrained in the box model. For the scenario with observed OVOCs constrained (red lines), we cannot determine the contribution of carbonyls with more carbon number to P(RO$_X$) because they are not possible to assign into specific individual species. Therefore, using the possible ranges of photolysis frequencies of carbonyls with more carbon number, we can obtain the possible widest variation range of P(RO$_X$). The red solid line represents the minimum OVOCs contribution and the dashed red line represents the maximum OVOCs contribution. The blue line is lower than the red solid line and red dashed line, indicating the simulated P(RO$_X$) without observed OVOCs constrained is

lower than that with observed OVOCs constrained for either minimum or maximum photolysis frequencies used.

**Line 343-347: As shown in Figure 3a, the simulation of the box model without observed OVOC species constrained (blue line in Figure 3a) underestimated P(RO$_X$) significantly compared to the scenario with all observed OVOC species constrained (red lines in Figure 3a).**

**Line 386-389: As shown in Figure 3b, compared to the scenario with observed photodegradable OVOCs species constrained in box model (red lines in Figure 3b), the scenario without constraining observed OVOCs (blue line in Figure 3b) underestimated peak value of P(O$_3$) by 15~38%.**

Technical Comments

L34: "comparable to"

Reply: Thank you. We have revised it.

L49: replace "hydroxide" with "hydrogen oxide"

Reply: Thank you. We have revised it.

L51: Recommend replacing "Photodegradable species…including" with "Photolysis of" and deleting "via photolysis reactions."

Reply: Thank you. We have revised it.

L139: PTR-ToF-MS

Reply: Thank you. We have revised it.

L202: "radicals"

Reply: Thank you. We have revised it.

L207: space after period

Reply: Thank you. We have revised it.

L270: delete "basically"

L271: "species and calculated photolysis frequencies derived from observed actinic flux."

Reply: Thank you. We have deleted this sentence and added another one "The simulated total P(RO$_X$) contains the contributions from all observed photodegradable species and several simulated OVOCs that was not measured such as glyoxal." to depict it more clearly.

L361: Start a new paragraph.

Reply: Thank you. We have revised it.

L424: Availability of model code?

Reply: Thank you. We have added it.

**Line 462-463: The observational data and model code used in this study are available from corresponding authors upon request (byuan@jnu.edu.cn).**

L543: Qu reference missing information.

Reply: Thank you. We have revised it.

L661: What is shading in Fig. 1?

Reply: Thank you. Lines and shading represent averages and standard deviations, respectively. We have explained it in the notation.

**Lines 706-708: Figure 1. The average diurnal variations of the concentrations of photodegradable OVOCs species during the field campaign in Guangzhou. Lines and shading represent averages and standard deviations, respectively.**

L667: ozonolysis is in this figure but never mentioned in the paper Supplement,

Reply: Many thanks. I have added a sentence to mention it.

**Line 310-311: Ozonolysis of alkenes play a minor role in P(RO$_X$).**

L57: Define ChL here.

Reply: Many thanks. I have defined ChL in lines 416-419 according to your suggestion.

**Line 413-416: ChL characterizes the number of iterations each RO$_X$ radical makes prior to termination. It is equal to the ratio between the radical recycling rate and primary production rate (or equivalently, termination rate), indicating the efficiency of radical propagation.**

Table S1: 4th column is a mixing ratio, not concentration. Also you may wish to include measurement uncertainty here (your choice).

Reply: Thank you. We have modified it.

**Lines 223-224: Table S1. Molecular formula, photolysis reactions, daytime average mixing ratio and photolysis frequencies of photodegradable species during the campaign.**

**References:**

Burkholder, J., Sander, S., Abbatt, J., Barker, J., Cappa, C., Crounse, J., Dibble, T., Huie, R., Kolb, C., and Kurylo, M.: Chemical kinetics and photochemical data for use in atmospheric studies; evaluation number 19, Pasadena, CA: Jet Propulsion Laboratory, National Aeronautics and Space …, 2020.

Griffith, S. M., Hansen, R., Dusanter, S., Michoud, V., Gilman, J., Kuster, W., Veres, P., Graus, M., de Gouw, J., and Roberts, J.: Measurements of hydroxyl and hydroperoxy radicals during CalNex-LA: Model comparisons and radical budgets, Journal of Geophysical Research: Atmospheres, 121, 4211-4232, 2016.

Jenkin, M. E., Saunders, S. M., and Pilling, M. J.: The tropospheric degradation of volatile organic compounds: a protocol for mechanism development, Atmos. Environ., 31, 81-104, 1997.

Qu, H., Wang, Y., Zhang, R., Liu, X., Huey, L. G., Sjostedt, S., Zeng, L., Lu, K., Wu, Y., and Shao, M.: Chemical Production of Oxygenated Volatile Organic Compounds Strongly Enhances Boundary-Layer Oxidation Chemistry and Ozone Production, Environmental Science & Technology, 55, 13718-13727, 2021.

Sekimoto, K., Li, S.-M., Yuan, B., Koss, A., Coggon, M., Warneke, C., and de Gouw, J.: Calculation of the sensitivity of proton-transfer-reaction mass spectrometry (PTR-MS) for organic trace gases using molecular properties, International Journal of Mass Spectrometry, 421, 71-94, 10.1016/j.ijms.2017.04.006, 2017.

Tan, Z., Fuchs, H., Lu, K., Hofzumahaus, A., Bohn, B., Broch, S., Dong, H., Gomm, S., Häseler, R., He, L., Holland, F., Li, X., Liu, Y., Lu, S., Rohrer, F., Shao, M., Wang, B., Wang, M., Wu, Y., Zeng, L., Zhang, Y., Wahner, A., and Zhang, Y.: Radical chemistry at a rural site (Wangdu) in the North China Plain: observation and model calculations of OH, HO2 and RO2 radicals, Atmos. Chem. Phys., 17, 663-690, 10.5194/acp-17-663-2017, 2017.

Tan, Z., Rohrer, F., Lu, K., Ma, X., Bohn, B., Broch, S., Dong, H., Fuchs, H., Gkatzelis, G. I., Hofzumahaus, A., Holland, F., Li, X., Liu, Y., Liu, Y., Novelli, A., Shao, M., Wang, H., Wu, Y., Zeng, L., Hu, M., Kiendler-Scharr, A., Wahner, A., and Zhang, Y.: Wintertime photochemistry in Beijing: observations of ROx radical concentrations in the North China Plain during the BEST-ONE campaign, Atmos. Chem. Phys., 18, 12391-12411, 10.5194/acp-18-12391-2018, 2018.

Tan, Z., Lu, K., Hofzumahaus, A., Fuchs, H., Bohn, B., Holland, F., Liu, Y., Rohrer, F., Shao, M., and Sun, K.: Experimental budgets of OH, HO 2, and RO 2 radicals and implications for ozone formation in the Pearl River Delta in China 2014, Atmospheric chemistry and physics, 19, 7129-7150, 2019a.

Tan, Z., Lu, K., Jiang, M., Su, R., Wang, H., Lou, S., Fu, Q., Zhai, C., Tan, Q., Yue, D., Chen, D., Wang, Z., Xie, S., Zeng, L., and Zhang, Y.: Daytime atmospheric oxidation capacity in four Chinese megacities during the photochemically polluted season: a case study based on box model simulation, Atmos. Chem. Phys., 19, 3493-3513, 10.5194/acp-19-3493-2019, 2019b.

Ye, C., Yuan, B., Lin, Y., Wang, Z., Hu, W., Li, T., Chen, W., Wu, C., Wang, C., Huang, S., Qi, J., Wang, B., Wang, C., Song, W., Wang, X., Zheng, E., Krechmer, J. E., Ye, P., Zhang, Z., Wang, X., Worsnop, D. R., and Shao, M.: Chemical characterization of oxygenated organic compounds in the gas phase and particle phase using iodide CIMS with FIGAERO in urban air, Atmospheric Chemistry and Physics, 21, 8455-8478, 10.5194/acp-21-8455-2021, 2021.

Young, C. J., Washenfelder, R. A., Roberts, J. M., Mielke, L. H., Osthoff, H. D., Tsai, C., Pikelnaya, O., Stutz, J., Veres, P. R., and Cochran, A. K.: Vertically resolved measurements of nighttime radical

reservoirs in Los Angeles and their contribution to the urban radical budget, Environmental science & technology, 46, 10965-10973, 2012.

---

## Author Response (AR2)

**Response to the editor and reviewers' comments**

Editor's comments

Dear authors: Please make the changes suggested by Reviewer #1. Additional grammatical suggestions are provided in the non-public comments to authors. I will leave it up to you as to whether the data/code are made publicly available as suggested by Reviewer #2.

Non-public comments to the Author:

While the manuscript is very clearly written and easy to follow, I think some grammatical improvements can be made. In addition to addressing the comments from the two reviewers, please replace the existing wording with the phrases suggested below. Many thanks !

John\

Line 27: "When evaluating their impact…"
Line 30: "…with a gas chromatography…" . Also, delete the word "technology".
Line 35: "… , we…"
Line 39: "… can contribute large fractions …"
Line 40: "which are comparable…"
Line 57: "… species are the largest…"
Line 75: "… due to limitations on available instrumentation …"
Line 85: "… heterogeneous uptake on aerosols…"
Line 100 (and other places): I think "OVOC species" is better.
Line 115: "… to determine the background…"
Line 140: "… lower than the concentration of …"
Line 160: "…because of its additional carbonyl functional group…"
Line 222: "… all correspond to radical formation channles, and do not include contributions from channels forming stable molecules."
Line 252: "… PTR-ToF-MS and GC-MS instruments to …"
Line 268: "… were contributed to by both…"
Line 291: "… with larger carbon number…"
Line 313: "… that reported that OVOCs contributed…"
Line 333: "… cross-sections and quantum yields…"
Line 350: "… concentrations was…"
Line 356: "… secondary sources…"
Line 451: "… can be measured by emerging online chemical …"

Reply: Many thanks! We have modified these grammar errors according to your suggestions.

Reviewer #2:

Minor correction:

Lines 293-297, this sentence is not clear. The readers cannot understand why the other OVOCs calculated by model simulations may lead to large uncertainties.

Reply: Many thanks. We have further explained it.

**Line 299-302: which may lead to large uncertainties. These uncertainties are likely, due to various possibility in modelling errors, including missing primary emissions of OVOCs (McDonald et al., 2018), unknown secondary sources of OVOCs (Bloss et al., 2005;Ji et al., 2017), heterogenous uptake on aerosols and unknown dilution and transport processes (Li et al., 2014).**

Line 332, "observation-determined $P(RO_X)$" might be typos.

Reply: Many thanks. We have deleted it.

The difference of the carbonyls' concentrations measured by the GC-MS and PTR-ToF-MS is very large (Fig. S1), and thus it is better to mention the uncertainty of OVOCs measurements in the text.

Reply: Many thanks. We have modified it accordingly.

**Line 36-41 in Supplement: The measurement results of the two instruments are generally similar (Figure S1). The differences of the two instruments for MVK+MACR, $C_3H_4O$ and $C_4H_8O$ are within 20%. However, acetone measured by GC-MS is 46% higher than that measured by PTR-ToF-MS. The differences between GC-MS and PTR-ToF-MS are acceptable, as uncertainties of OVOCs measurements of GC-MS and PTR-ToF-MS are in the range of 20-30%.**